# *Coxiella burnetii* in free-living feral pigs *(Sus scrofa)* in Brazil

Jorlan Fernandes[1]/[+], Wania Guimarães dos Santos[1], Matheus Ribeiro da Silva Assis[1], Martha Lima Brandão[2], Marcione Brito de Oliveira[3,4], Jairo Dias Barreira[5], Dominique Elvira Souza Freitas[1], José Luís Passos Cordeiro[6], Luiz Flamarion Barbosa de Oliveira[4], Elba Regina Sampaio de Lemos[1]

[1]Fundação Oswaldo Cruz-Fiocruz, Instituto Oswaldo Cruz, Laboratório de Hantaviroses e Rickettsioses, Rio de Janeiro, RJ, Brasil
[2]Fundação Oswaldo Cruz-Fiocruz, Vice-Presidência Produção e Inovação em Saúde, Rio de Janeiro, RJ, Brasil
[3]Fundação Oswaldo Cruz-Fiocruz, Instituto Oswaldo Cruz, Laboratório de Biologia e Parasitologia de Mamíferos Silvestres Reservatórios, Rio de Janeiro, RJ, Brasil
[4]Universidade Federal do Rio de Janeiro, Museu Nacional, Departamento de Vertebrados, Mastozoologia, Rio de Janeiro, RJ, Brasil
[5]Universidade Federal do Estado do Rio de Janeiro, Departamento de Microbiologia e Parasitologia, Rio de Janeiro, RJ, Brasil
[6]Fundação Oswaldo Cruz-Fiocruz, Fortaleza, CE, Brasil

**BACKGROUND** *Coxiella burnetii*, the etiological agent of coxiellosis in animals and Q fever in humans, is a zoonotic pathogen of global relevance that can infect a wide range of species. Although several domestic and wild animals are involved in the natural cycle, the role of wildlife hosts remains poorly understood.

**OBJECTIVES** Our study aimed to investigate the presence of *C. burnetii* in feral pigs hunted in Brazilian Pantanal wetland.

**METHODS** In this study, 36 free-living feral pigs legally hunted in Mato Grosso State, Brazil, were sampled. Sera were tested by enzyme-linked immunosorbent assay (ELISA), and spleen, liver, sera and tick samples were analysed by polymerase chain reaction (PCR).

**FINDINGS** Serological evidence of exposure was detected in 22.2% [8/36; 95% confidence interval (CI): 11.7% - 38.1%], while *C. burnetii* DNA was found in one spleen sample (1/36 - 2.8%; 95% CI: 0.1% - 14.5%). Only *Coxiella*-like endosymbiont was detected in *Amblyomma sculptum* ticks (9/23 - 39.13%; 95% CI: 22.2% - 59.2%).

**MAIN CONCLUSIONS** These results represent the first detection of *C. burnetii* in free-living feral pigs in Brazil and suggest potential exposure of this invasive mammal species to the pathogen. The findings underscore the need for broader surveillance of *C. burnetii* at the wildlife-livestock-human interface in Brazil.

Key words: *Coxiella burnetii* - feral pigs - wild boars - ticks - endosymbiont - Q fever - coxiellosis

*Coxiella burnetii* is a Gram-negative bacterium that causes disease in humans (Q fever) and animals (coxiellosis). Q fever is usually asymptomatic or presents with mild flu-like symptoms, although infection may be severe, especially in individuals with underlying conditions such as immunocompromised and cardiac valve defects, which can develop serious complications like pneumonia, endocarditis and hepatitis.[1,2,3] Coxiellosis causes similar clinical outcomes and pathologies in wild and domestic animals, especially in pregnant female, that may experience reproductive problems such as placentitis, abortion and low birth weight.[4,5]

Various mammals, birds and reptiles have been reported to be infected with *C. burnetii*, including both domestic species and wildlife.[6,7] The primary route of infection are through inhalation of aerosols or dust contaminated with placental materials, faeces, or vaginal secretions from infected animals.[1,5,4] Ticks are thought to play a pivotal role transmitting the pathogen in livestock and wildlife mammals by feeding on an infected host and subsequently passing it to the next host during later feedings.[6,7,8,9] Also, feeding behavioural patterns could modulate exposure to *C. burnetii* in humans, through the consumption of raw meat and milk, and in wild animals indicating that predator-prey relationships, presumably by ingestion of these prey species.[1,6,7,10,11]

Wildlife species serve as force multipliers, thereby spreading the pathogen from infected environments to diverse and remote habitats. Furthermore, there is an increasing overlap between habitats of wild animals and human residential areas due to deforestation, land development, and food scarcity.[12,13,14,15] This indicates that wild animals have the potential to contribute to the spread of *C. burnetii* among humans, domestic and production animals.[6,7,10,11] Among wild animals, feral pigs and wild boars (*Sus scrofa*) are known to serve as reservoirs for *C. burnetii* in various countries.[16-23]

---

**doi:** 10.1590/0074-02760250284

**Financial support:** FAPERJ (grant number E-26/200.022/2024), CNPq (grants numbers 400713/2013-6 and 303024/2019-4).

**+ Corresponding author:** jorlan@ioc.fiocruz.br | ⓘ https://orcid.org/0000-0002-5039-0604

**Handling editor:** Adeilton Alves Brandão | ⓘ https://orcid.org/0000-0001-5877-607X

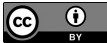

Feral pigs are distributed worldwide and have an extensive living range and movement patterns across the world.[16,24] In Brazil, first records of feral pigs were reported in the 1960s and increased in the late 1980's and early 1990's, showing an accelerated increase in the last 30 years, and found in all six Brazilian biomes.[25,26,27] Feral pigs, also known as "porco monteiro", were introduced into the Pantanal biome (wetlands) as domestic pigs, that later become feral during the Paraguayan War (1865-1870), when farms were abandoned, and the animals escaped into the wild.[28] Currently, it is estimated that there are approximately ten thousand feral pig herds widely distributed throughout the Pantanal's landscapes where they apparently contribute to the conservation status of wildlife in the region and are of great importance to Pantanal culture, as there is a clear preference for hunting of feral pigs.[25,26,27,28,29] Therefore, our study aimed to investigate the presence of *C. burnetii* in feral pigs legally hunted in Mato Grosso State, Brazil.

## MATERIALS AND METHODS

A cross-sectional study was conducted between November 2014 and October 2015, in the municipality of Barão de Melgaço, Mato Grosso State in the Northeastern Brazilian Pantanal, on a traditional cattle ranch (16º53'38.25''S, 55º54'24.98''W; Figure).[30] Samples were obtained from freshly hunted animals, provided by local residents and ranch workers engaged in traditional subsistence hunting commonly practiced in the region.[29] No animals were hunted or handled alive for research purposes, and all collections complied with ethical and legal regulations. The project team did not handle live animals consequently ethical approval was not deemed necessary by Fiocruz Animal Use Ethics Committee (P.11/2025.1), SISBIO license 49647-1. From each freshly hunted individual, blood, tissue fragments (liver and spleen), and ticks were collected after external inspection of the animals' bodies. Blood (2-10 mL) were sampled in tubes without anticoagulants and subsequently centrifuged, tissues and ticks were stored individually and kept frozen at -20ºC until laboratory processing.

Sera samples were tested using an indirect enzyme-linked immunosorbent assay (ELISA) for the detection of phases I and II antibodies directed against *C. burnetii* for the presence of antibodies to *C. burnetii* in multiple species (ID Screen® Q fever Indirect Multi-species; IDvet, Montpellier, France), in accordance with the manufacturer's recommendations. Optical densities (OD) of the tested samples and positive and negative controls were measured by an ELISA plate reader at 450 nm. The OD ratio of the sample and positive control (S/P) was calculated for each sample as follows: [(ODsample - ODnegative) / (ODpositive - ODnegative)] × 100. Ratios were stratified as three different rising categories: samples with S/P < 40% were considered negative, samples with S/P between 40% and 50% were considered doubtful, samples with S/P > 50% were considered positive, according to the manufacturer's recommendations. Any serum sample that was initially classified as "doubtful" was retested and, if resulting doubtful again, it was then considered as negative.

DNA was extracted from blood, tissue samples and tick specimens collected from feral pigs using QIAamp DNA mini kit (Qiagen, Valencia, CA, USA), following the manufacturers' instructions. Ticks were individually washed twice with distilled water to remove any adherent host material, and engorged ticks were dissected using separate sterile forceps and scalpel blades to separate head and legs that were used for DNA extraction. Subsequently, feral pigs' blood and tissue's DNA were submitted to a conventional polymerase chain reaction (PCR) targeting the endogenous mammalian glyceraldehyde-3-phosphate dehydrogenase (GAPDH) gene, to ascertain the presence of amplifiable DNA and then submitted to PCR assays.[31] Bacterial DNA was detected using *Coxiella* spp. and *C. burnetii*-specific primers, designed to amplify the 16S rRNA and the repetitive element IS1111 associated with the transposase gene, respectively. Controls and conditions for Nested-PCR standardised previously.[32,33]

Amplified products were separated by electrophoresis in 1.5% agarose gels stained with Gel Red Nucleic Acid Gel Stain™ 10000× in DMSO (Biotium, Hayward, CA, USA) under 100 V/150 mA electric current for 60 min. The products of the PCR assays were purified using the ExoSAP-IT PCR Product Cleanup Reagent enzyme (Applied Biosystems, Foster City, CA, USA) and sequenced using the BigDye™ Terminator v3.1 Cycle Sequencing Kit (Thermo Fisher Scientific™, Waltham, MA, USA) and the ABI PRISM 310 DNA Analyzer (Applied Biosystems™, Foster City, CA, USA).

Taxonomic identification of the collected ticks was performed using a stereomicroscope (Olympus Corporation, Tokyo, Japan) and following previously described taxonomic keys for adult ticks.[34,35] Ticks were also identified by molecular analysis and for this purpose, after nucleic acid extraction DNA samples were tested by PCR assay targeting a portion of the tick mitochondrial 16S rRNA gene, as previously described.[36]

## RESULTS

A total of 36 individuals were analysed for *C. burnetii* infection: 36 blood samples, 26 spleen and liver fragments. Antibodies to *C. burnetii* were detected in 22.2% (8/36; 95% CI: 11.7% - 38.1%) feral pigs (five males and three females) [Table and Supplementary data (Table). *C. burnetii*' IS1111 partial gene was amplified in one spleen sample 2.8% (1/36; 95% CI: 0.1-14.5) (GenBank PX501780) of a non-reactive male feral pig hunted in 2014, showing identity values higher than 99% with *C. burnetii* (GenBank MN868471).

Ticks were collected, by convenience, from four out of 36 (11.11%) animals. All 23 ticks (nine males and 14 females) collected were identified as adults *Amblyomma sculptum* specimens (GenBank PX455748 - PX455750). In the PCR-based screening assays for Coxiellaceae agents based on the 16S rRNA gene, 39.13% (9/23; 95% CI: 22.2% - 59.2%; GenBank PX455199 - PX455206) of the ticks (five males and four females) were positive showing high identity with *Coxiella*-like endosymbiont (CLE) from *A. sculptum* collected in Brazil (GenBank CP033868) [Supplementary data (Table)].

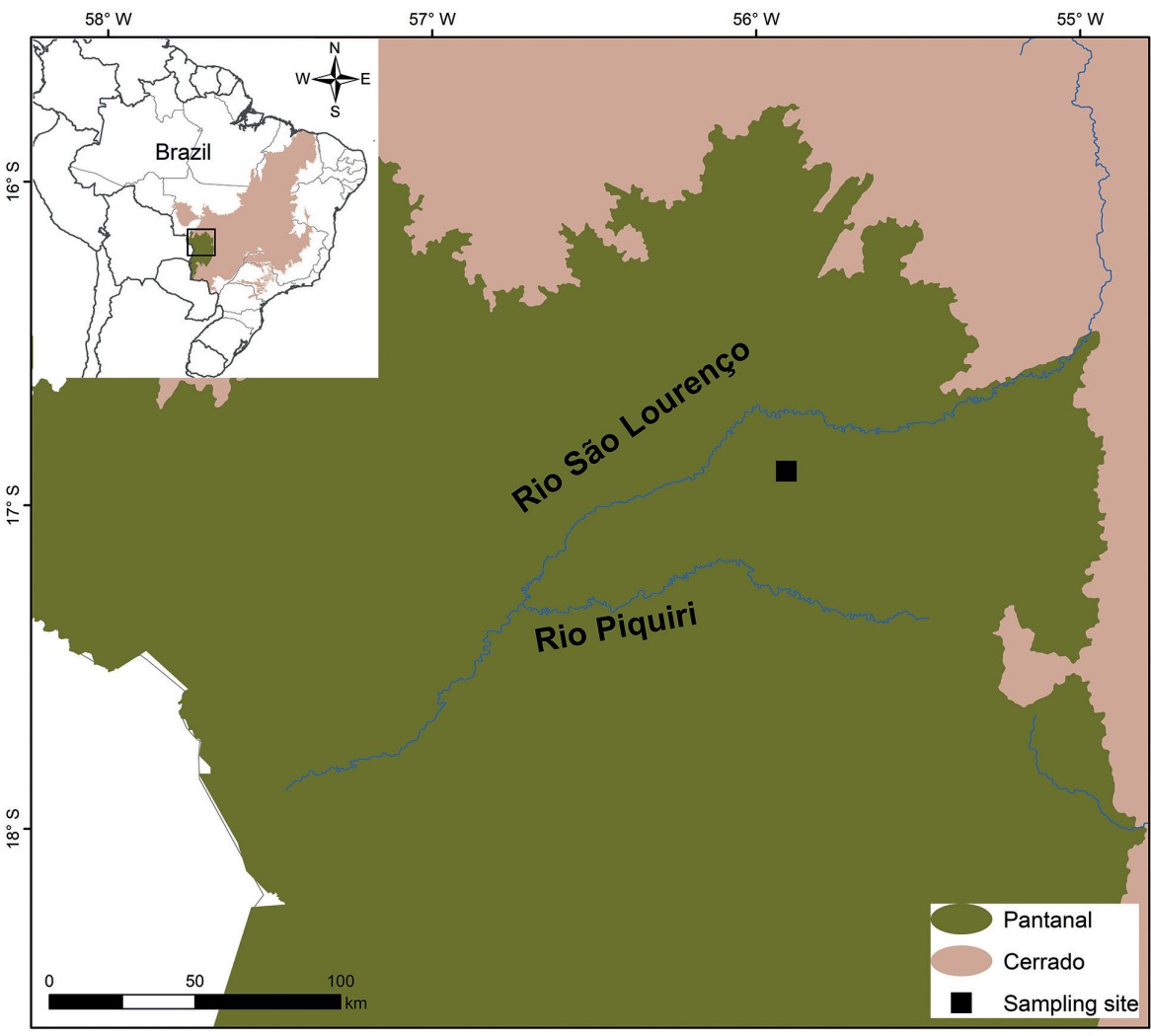

Map showing the data collection site of the feral pigs (*Sus scrofa*) on a traditional cattle ranch in the municipality of Barão de Melgaço, Mato Grosso, in the northeastern Pantanal, Brazil. Source: IBGE Biomes (Available from: https://www.ibge.gov.br/geociencias/cartas-e-mapas/informacoes-ambientais/15842-biomas.html).

## TABLE

Distribution of characteristics, prevalence of *Coxiella burnetii* infection for free-living feral pigs (*Sus scrofa*), Mato Grosso State, Brazil

| Variable | N (%) | Seropositivity (%) | PCR (%) |
|---|---|---|---|
| Sex | | | |
| Female | 14 (38.9) | 3 (21.4) | 0 (0.0) |
| Male | 22 (61.1) | 5 (22.7) | 1 (4.5) |
| Year | | | |
| 2014 | 23 (63.9) | 5 (21.74) | 1 (4.3) |
| 2015 | 13 (36.1) | 3 (23.1) | 0 (0.0) |
| Tick infestation | | | |
| Yes | 4 (11.1) | 1 (25.0) | 1 (25.0) |
| No | 32 (88.8) | 7 (21.9) | 0 (0.0) |
| Total | 36 (100.0) | 8 (22.22) | 1 (2.8) |

## DISCUSSION

*Coxiella burnetii* is an example of a pathogen that needs the One Health perspective considering the environmental risk associated with domestic, production and wild mammals, particularly in regions of nature, where the human population is in close contact with agriculture and, consequently, a higher proximity with livestock and wildlife. This is the scenario of Brazilian wetlands (Pantanal biome). The Pantanal biome is a large floodplain in South America that covers 160,000 km². In this area, extensive livestock production is the main economic activity, characterised by livestock-wildlife interface.[37,38,39] Cattle and horses share the same habitats as the abundant wildlife including feral pigs, and white-lipped peccary (*Tayassu pecari*), Southern coati (*Nasua nasua*), the ocelot (*Leopardus pardalis*) and the crab-eating fox (*Cerdocyon thous*).[39,40,41] In this complex scenario *C. burnetii* represent a risk for humans and animals.

To date, only two studies have indicated the circulation of *C. burnetii* in Brazilian wetlands, in beef cattle (1.0% - 2/200) and in neotropical free-living brocket deer (*Mazama gouazoubira*) at lower rates (6.2% - 2/32) than the one described here (22.22%).[42,43]

The seropositivity found in our study is also higher than those reported previously for wild boars populations sampled in South Korea (14.6% - 142/975), Czech Republic (6% - 2/32), Spain (5.48% - 4/73), Portugal (1.1% - 4/358) and United States of America (0.47% - 5/1.063).[16,17,18,22,23] No association was found between seroprevalence and sex, age or clinical signs, since animals showed no signs of disease. Coxiellosis causes similar clinical outcomes and pathologies in wild animals as it does in domestic animals and livestock, usually with no clear signs of infection, thus seroepidemiological studies that show the presence of infection are important in disease control, since wild ungulates can transmit the agent even while providing a seronegative result.[18,44] In fact, here *C. burnetii* DNA was amplified from a spleen sample of a seronegative animal (2.80%) — indicating active infection before antibody production (seroconversion), or immune evasion leading to latency and localisation in the spleen — in a lower prevalence than described in wild boars from Spain (4.3% - 4/93) and slightly higher than the one reported in Italy (1.6% -1/63).[19,21] The low PCR detection rate in means that the epidemiological role of feral pigs as an active reservoir should be interpreted with caution.

Ticks are known to play a significant role in the transmission of *C. burnetii* in wildlife populations, and they thrive in environments with abundant moisture and tall grass, which provide shade from the harsh sun.[7,8,9,16,45] The presently observed high incidence of 39.13% of CLE, closely related but genetically distinct to *C. burnetii*, in *A. sculptum* ticks from feral pigs has also been demonstrated in a wide variety of Ixodidae ticks including *Amblyomma* spp.[6,7,46] CLE are almost exclusively confined to ticks and pose a much lower infection risk to vertebrates, but has been commonly misidentified as *C. burnetii*. Based on the presence of IS1111 in both *C. burnetii* and the CLE, our group developed a nested-PCR assay, considering that IS1111 sequences in endosymbionts revealed to be genetically divergent, providing a specific *C. burnetii* detection, which resulted on a feral pig positive-PCR, but with no detection on *A. sculptum* ticks, including those collected from the PCR-positive pig.[6,33]

Although *A. sculptum* ticks have previously been found to be positive for *C. burnetii*, recent studies in feral pigs and associated ticks from São Paulo State, Brazil, no blood or tick samples showed to be positive in the qPCR for *C. burnetii* based on the IS1111 gene.[33,45] Indicating that ticks may not be the main source of infection for feral pigs in Brazil. *C. burnetii* is probably far more frequently transmitted through the airborne route or consumption than through tick bites. Feral pigs are omnivores, meaning they have a diverse diet, opportunistically exploiting food resources according to availability during different seasons. Livestock farming is believed to buffer seasonal environmental fluctuations for feral pigs, providing artificial water sources, cattle carcasses for food. Thus, there are reports of predation of newborn calves and lambs, all of which could result in *C. burnetii* transmission.[28,30] Additional studies involving feral pigs are needed to allow a better understanding of prevalence and transmission routes in Brazilian wetland ecosystems.

Considering that *C. burnetii* infections in wild mammals have been associated with infertility, abortion, stillbirth, endometritis, and mastitis, the detection of *C. burnetii* presented in this study should serve as a warning to professionals working in conservation programs for sensitive species, anticipating prey-to-predator spillover events, which have had significant implications.[5,11] As an integral part of the trophic structure of Pantanal communities, feral pigs are important prey for predators in the food chain, such as the jaguar (*Panthera onca*), is important to evaluate the role of trophic webs as determining factor in *C. burnetii* enzootic cycle in a natural ecosystem.[28,47]

Feral pigs are a source of meat and fat, providing food security for the Pantanal community; hunting and castration activities strengthen social bonds of the local population. Handling infected animals can be an important source of *C. burnetii* infection for human and domestic animals like dogs. In Queensland almost one in five pig-hunting dogs (18.3%; 19/104) were seropositive to *C. burnetii*. One of the highest recorded in Australia, indicating that feral pigs and dogs could potentially be sources of *C. burnetii*.[48] Although the zoonotic disease risk posed by feral pigs is poorly understood, our findings indicate that hunters should be aware of the risk of exposure to Q fever during hunts.

This study provides the first detection of *C. burnetii* in free-living feral pigs in Brazil, indicating that this invasive and exotic species may be exposed to the pathogen. While the limited sample size and the small number of PCR-positive animals require cautious interpretation, the results highlight the importance of further investigations with larger sample sizes to assess the epidemiological role of feral pigs in the transmission of *C. burnetii*. These findings support the need for continuous One Health surveillance at the wildlife-livestock-human interface.

## ACKNOWLEDGEMENTS

To Gustavo Staut for field support during the field work and for granting permission to work on the Sta. Lucia Ranch and the Genomic Platform — DNA Sequencing — RPT01A (Rede de Plataformas Tecnológicas - FIOCRUZ) for technical support.

## AUTHORS' CONTRIBUTION

JF, WGS, MRSA and DESF performed lab experiments draft and reviewed the manuscript; MLB, MBO and JLPC conducted field studies and reviewed the manuscript; JDB performed tick taxonomy and reviewed the manuscripts; JLPC, LFBO and ERSL coordinated the resources and reviewed the manuscript.

## DATA AVAILABILITY

The sequences generated during the current study are available in GenBank under the following accession numbers: PX501780, PX455199 - PX455206 and PX455748 - PX455750.

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

# OPEN PEER REVIEW

Memórias do IOC thanks the anonymous reviewers for their contribution to the peer review of this work.

## FIRST REVIEW ROUND

REVIEWERS' COMMENTS

### REVIEWER #1

The manuscript under review describes the first detection of Coxiella burnetii in wild pigs (Sus scrofa) in Brazil, a finding of great relevance to One Health in the Pantanal. The study is methodologically sound, especially in its use of specific molecular techniques to differentiate C. burnetii from related endosymbionts. The originality of the work is very high, since this study represents the first detection of C. burnetii in free-living wild pigs in Brazil.

Below are the aspects that need improvement:

The abstract is well structured and informative, covering objectives, methods, results, and conclusion. However, its adequacy can be improved through minor adjustments that reinforce the main narrative. The abstract states that 'Only Coxiella-like endosymbiont was detected in Amblyomma sculptum ticks.' Include the percentage of infection.

The abstract states that the spleen, liver, and ticks were analysed by PCR. The results indicate that only 26 of the 36 spleen and liver samples were analysed. Although the difference is small, we suggest a slight rewriting of the abstract to indicate 'Sera were tested by ELISA, and tissues (26 spleen and liver fragments) and tick samples...'

The study is pioneering, but the authors acknowledge the limitation of the small number of PCR-positive animals (only 1/36; 2.8%). The importance of the molecular finding (the PCR-positive animal was seronegative) suggests an active infection. However, it is necessary to explicitly emphasise, at the end of the Introduction or at the beginning of the discussion, that this low PCR detection rate means that the epidemiological role as an active reservoir should be interpreted with statistical caution (given the wide confidence interval).

The point about the seronegative but PCR-positive animal is essential. The discussion should be expanded to explore possible biological and epidemiological explanations for this observation in the context of wild ungulates (e.g., acute infection prior to seroconversion, failure of immune response, or latent infection and localisation of the bacterium in the spleen).

It is essential that the footnote in Supplementary Table 1 on '*only blood was collected*' be explicit so that the reader immediately understands why tissue analysis (PCR) was performed on only 26 animals, rather than on all 36 animals.

### REVIEWER #2

This is a valuable and important preliminary research study. It successfully achieved its primary goal of providing the first documented evidence of Coxiella burnetii exposure and infection in feral pigs in Brazil. The study was well-designed as a field-based survey, used appropriate and validated diagnostic methods, and thoughtfully contextualized its findings within the One Health framework. However, its main limitations are related to its scale and the inherent constraints of a cross-sectional study, which the authors have appropriately acknowledged. This study fills a critical knowledge gap by demonstrating that an abundant invasive species, such as feral pigs, is exposed to and can carry C. burnetii in a key Brazilian biome is a significant finding. This raises important concerns regarding wildlife health, livestock biosecurity, and public health. The authors highlight the results in the context of One Health. They connect it to livestock by mentioning the extensive cattle ranching in the Pantanal, wildlife by discussing potential spillover to predators like jaguars, and human health by explicitly warning hunters and local communities about the risk of Q fever. The paper has a good methodology, and using diagnostic tools such as serology (ELISA) and molecular biology (PCR) provides a more complete picture. Serological tests showed historical exposure, while PCR confirmed active infection. The finding of a PCR-positive, seronegative animal is a classic and important demonstration of an acute infection. The use of the IS1111 gene for C. burnetii-specific PCR is the gold standard and crucial for differentiating the pathogen from non-pathogenic Coxiella-like endosymbionts (CLE) found in ticks. This adds high confidence to the single positive detection. The ethical aspects are important because they used animals previously hunted. It provides scientific data without additional ethical costs and aligns the research with local practices.

The discussion effectively compares the findings (both seroprevalence and PCR prevalence) with those of other studies worldwide, providing good context. Speculation on potential transmission routes (airborne, consumption of livestock carcasses/afterbirth) is well explained.

The authors have franked the limitations of the study. This is mainly due to the limited sample size. This is the most significant limitation explicitly noted by the authors. With only 36 animals, the confidence intervals for both seroprevalence (11.7%–38.1%) and PCR prevalence (0.1% – 14.5%) were very wide. This means that the "true" infection rate in the population could be anywhere within these broad ranges and subdiagnostic. The findings may

indicate the circulation of Coxiella, but cannot be used to estimate its precise prevalence in the wider feral pig population. The work is a cross-sectional design that shows a snapshot in time, and the study can demonstrate that infection is present, but cannot answer dynamic questions about how the pathogen is maintained and transmitted within the ecosystem. It establishes an association, not causation. This is not a poor design and is not the author's responsibility. Hence, only one animal was PCR-positive out of the 36 tested. While this is a crucial finding, it limits the ability to perform further analyses, such as genetic sequencing, to compare the strain with those found in livestock or humans in the region. A larger sample size may yield more positive tissues for comparative studies. Ticks were collected from only 4 four of the 36 animals (11%). This convenience sampling means that the data obtained from A. sculptum ticks are not representative of the tick burden in the entire study population. The statement that ticks may not be the main source of infection is uncautious because it is based on an extremely limited tick population. The authors should be more cautious about this. While the discussion of alternative transmission routes (airborne, dietary) is appropriate, the study design does not allow for testing these hypotheses. Again, the authors must be cautious regarding the limitations of the study.

In conclusion, this article is very interesting, especially for the intelligent way to obtain Coxiella from hunted animals. The best contribution is to put a new host species and a critical region on the map for C. burnetii. The investigators proved that C. burnetii is circulating in feral pigs in the Pantanal. It effectively uses limited data to build a compelling case for why this matters and why larger, more longitudinal studies are urgently needed to understand the epidemiology and risk at this dynamic wildlife-livestock-human interface.

## AUTHORS' RESPONSE TO THE REVIEWERS

We would like to thank the Editor and the Reviewers for their careful evaluation of our manuscript and for the constructive comments and suggestions. We believe that these comments have significantly improved the quality and clarity of our manuscript.

Reviewer: 1

The manuscript under review describes the first detection of Coxiella burnetii in wild pigs (Sus scrofa) in Brazil, a finding of great relevance to One Health in the Pantanal. The study is methodologically sound, especially in its use of specific molecular techniques to differentiate C. burnetii from related endosymbionts. The originality of the work is very high, since this study represents the first detection of C. burnetii in free-living wild pigs in Brazil.

Below are the aspects that need improvement:

The abstract is well structured and informative, covering objectives, methods, results, and conclusion. However, its adequacy can be improved through minor adjustments that reinforce the main narrative. The abstract states that 'Only Coxiella-like endosymbiont was detected in Amblyomma sculptum ticks.' Include the percentage of infection.

Response: Thank you for the suggestion. We have included the percentage of infection at the end of the sentence, which now reads: "Only Coxiella-like endosymbiont was detected in Amblyomma sculptum ticks (9/23 - 39,13%; 95% CI: 22,2% – 59,2%)".

The abstract states that the spleen, liver, and ticks were analysed by PCR. The results indicate that only 26 of the 36 spleen and liver samples were analysed. Although the difference is small, we suggest a slight rewriting of the abstract to indicate 'Sera were tested by ELISA, and tissues (26 spleen and liver fragments) and tick samples...'

Response: Thank you for pointing this out. We understand the potential confusion. In fact, serum samples were also analysed by PCR. We therefore rephrased the sentence in the abstract as follows: "Sera were tested by ELISA, and spleen, liver, sera and tick samples were analyzed by PCR."

We also added the total number of tested animals in the following sentence: "Serological evidence of exposure was detected in 22.2% (8/36; 95% CI: 11,7% – 38,1%), while C. burnetii DNA was found in one spleen sample (1/36 - 2.8%; 95% CI: 0.1% – 14.5%)."

The study is pioneering, but the authors acknowledge the limitation of the small number of PCR-positive animals (only 1/36; 2.8%). The importance of the molecular finding (the PCR-positive animal was seronegative) suggests an active infection. However, it is necessary to explicitly emphasise, at the end of the Introduction or at the beginning of the discussion, that this low PCR detection rate means that the epidemiological role as an active reservoir should be interpreted with statistical caution (given the wide confidence interval).

Response: Thank you for the suggestion. We have added the following sentence in the discussion, in the context of the PCR positive:

The low PCR detection rate in means that the epidemiological role of feral pigs as an active reservoir should be interpreted with caution."

The point about the seronegative but PCR-positive animal is essential. The discussion should be expanded to explore possible biological and epidemiological explanations for this observation in the context of wild ungulates (e.g., acute infection prior to seroconversion, failure of immune response, or latent infection and localisation of the bacterium in the spleen).

Response: We agree and thank the reviewer for this important point. We have added the following sentence to the Discussion

"In fact, here C. burnetii DNA was amplified from a spleen sample of a seronegative animal (2,80%) - indicating active infection before antibody production (seroconversion), or immune evasion leading to latency and localization in the spleen - in a lower prevalence than described in wild boars from Spain (4.3% - 4/93) and slightly higher than the one reported in Italy (1.6% -1/63). (19,21)"

It is essential that the footnote in Supplementary Table 1 on '*only blood was collected*' be explicit so that the reader immediately understands why tissue analysis (PCR) was performed on only 26 animals, rather than on all 36 animals.

Response: Thank you for this suggestion. To improve clarity, we revised the title of Supplementary Table 1 as follows:

"Suppl Table 1. Screening for Coxiella burnetii antibodies and DNA in 36 sera samples and 26 tissues of free-ranging feral pigs (Sus scrofa) and associated 23 ticks from Mato Grosso state, Brazil."

Reviewer: 2

This is a valuable and important preliminary research study. It successfully achieved its primary goal of providing the first documented evidence of Coxiella burnetii exposure and infection in feral pigs in Brazil. The study was well-designed as a field-based survey, used appropriate and validated diagnostic methods, and thoughtfully contextualized its findings within the One Health framework. However, its main limitations are related to its scale and the inherent constraints of a cross-sectional study, which the authors have appropriately acknowledged. This study fills a critical knowledge gap by demonstrating that an abundant invasive species, such as feral pigs, is exposed to and can carry C. burnetii in a key Brazilian biome is a significant finding. This raises important concerns regarding wildlife health, livestock biosecurity, and public health. The authors highlight the results in the context of One Health. They connect it to livestock by mentioning the extensive cattle ranching in the Pantanal, wildlife by discussing potential spillover to predators like jaguars, and human health by explicitly warning hunters and local communities about the risk of Q fever. The paper has a good methodology, and using diagnostic tools such as serology (ELISA) and molecular biology (PCR) provides a more complete picture. Serological tests showed historical exposure, while PCR confirmed active infection. The finding of a PCR-positive, seronegative animal is a classic and important demonstration of an acute infection. The use of the IS1111 gene for C. burnetii-specific PCR is the gold standard and crucial for differentiating the pathogen from non-pathogenic Coxiella-like endosymbionts (CLE) found in ticks. This adds high confidence to the single positive detection. The ethical aspects are important because they used animals previously hunted. It provides scientific data without additional ethical costs and aligns the research with local practices.

The discussion effectively compares the findings (both seroprevalence and PCR prevalence) with those of other studies worldwide, providing good context. Speculation on potential transmission routes (airborne, consumption of livestock carcasses/afterbirth) is well explained.

The authors have franked the limitations of the study. This is mainly due to the limited sample size. This is the most significant limitation explicitly noted by the authors. With only 36 animals, the confidence intervals for both seroprevalence (11.7%–38.1%) and PCR prevalence (0.1% – 14.5%) were very wide. This means that the "true" infection rate in the population could be anywhere within these broad ranges and subdiagnostic. The findings may indicate the circulation of Coxiella, but cannot be used to estimate its precise prevalence in the wider feral pig population. The work is a cross-sectional design that shows a snapshot in time, and the study can demonstrate that infection is present, but cannot answer dynamic questions about how the pathogen is maintained and transmitted within the ecosystem. It establishes an association, not causation. This is not a poor design and is not the author's responsibility. Hence, only one animal was PCR-positive out of the 36 tested. While this is a crucial finding, it limits the ability to perform further analyses, such as genetic sequencing, to compare the strain with those found in livestock or humans in the region. A larger sample size may yield more positive tissues for comparative studies. Ticks were collected from only 4 four of the 36 animals (11%). This convenience sampling means that the data obtained from A. sculptum ticks are not representative of the tick burden in the entire study population. The statement that ticks may not be the main source of infection is uncautious because it is based on an extremely limited tick population. The authors should be more cautious about this.

Response: Thank you for raising this important concern. We fully agree with the limitations highlighted. We did not intend to rule out the role of ticks as a source of infection. Based on our findings and previously published studies in Brazil, we revised the Discussion to adopt a more cautious interpretation. The revised text now reads: "Although A. sculptum ticks have previously been found to be positive for C. burnetii, recent studies in feral pigs and associated ticks from São Paulo state - Brazil, no blood or tick samples showed to be positive in the qPCR for C. burnetii based on the IS1111 gene. (33,45) Indicating that ticks may not be the main source of infection for feral pigs in Brazil. C. burnetii is probably far more frequently transmitted through the airborne route or consumption than through tick bites."

While the discussion of alternative transmission routes (airborne, dietary) is appropriate, the study design does not allow for testing these hypotheses. Again, the authors must be cautious regarding the limitations of the study.

Response: We agree and appreciate this observation. To address this limitation, we added the following sentence to the Discussion:

"Additional studies involving feral pigs are needed to allow a better understanding of prevalence and transmission routes in Brazilian wetland ecosystems."

## SECOND REVIEW ROUND

REVIEWERS' COMMENTS

### REVIEWER #1

The revised version of the manuscript has been carefully evaluated. The authors have adequately addressed all reviewers' comments, and the quality of the manuscript has been significantly improved.

### REVIEWER #2

I have no comments; the authors followed the referees' recommendations.

