## [Reviewer Report · FIRST REVIEW ROUND - REVIEWERS COMMENTS]

## REVIEWER #1

The manuscript under review describes the first detection of *Coxiella burnetii* in wild pigs (*Sus scrofa*) in Brazil, a finding of great relevance to One Health in the Pantanal.

The study is methodologically sound, especially in its use of specific molecular techniques to differentiate *C. burnetii* from related endosymbionts.

The originality of the work is very high, since this study represents the first detection of *C. burnetii* in free-living wild pigs in Brazil.

Below are the aspects that need improvement:

The abstract is well structured and informative, covering objectives, methods, results, and conclusion.

However, its adequacy can be improved through minor adjustments that reinforce the main narrative.

The abstract states that ‘Only *Coxiella*-like endosymbiont was detected in *Amblyomma sculptum* ticks.’ Include the percentage of infection.

The abstract states that the spleen, liver, and ticks were analysed by PCR.

The results indicate that only 26 of the 36 spleen and liver samples were analysed.

Although the difference is small, we suggest a slight rewriting of the abstract to indicate ‘Sera were tested by ELISA, and tissues (26 spleen and liver fragments) and tick samples...’

The study is pioneering, but the authors acknowledge the limitation of the small number of PCR-positive animals (only 1/36; 2.8%).

The importance of the molecular finding (the PCR-positive animal was seronegative) suggests an active infection.

However, it is necessary to explicitly emphasise, at the end of the Introduction or at the beginning of the discussion, that this low PCR detection rate means that the epidemiological role as an active reservoir should be interpreted with statistical caution (given the wide confidence interval).

The point about the seronegative but PCR-positive animal is essential.

The discussion should be expanded to explore possible biological and epidemiological explanations for this observation in the context of wild ungulates (e.g., acute infection prior to seroconversion, failure of immune response, or latent infection and localisation of the bacterium in the spleen).

It is essential that the footnote in Supplementary Table 1 on ‘*only blood was collected*’ be explicit so that the reader immediately understands why tissue analysis (PCR) was performed on only 26 animals, rather than on all 36 animals.

## REVIEWER #2

This is a valuable and important preliminary research study. It successfully achieved its primary goal of providing the first documented evidence of *Coxiella burnetii* exposure and infection in feral pigs in Brazil.

The study was well-designed as a field-based survey, used appropriate and validated diagnostic methods, and thoughtfully contextualized its findings within the One Health framework.

However, its main limitations are related to its scale and the inherent constraints of a cross-sectional study, which the authors have appropriately acknowledged.

This study fills a critical knowledge gap by demonstrating that an abundant invasive species, such as feral pigs, is exposed to and can carry *C. burnetii* in a key Brazilian biome is a significant finding.

This raises important concerns regarding wildlife health, livestock biosecurity, and public health.

The authors highlight the results in the context of One Health.

They connect it to livestock by mentioning the extensive cattle ranching in the Pantanal, wildlife by discussing potential spillover to predators like jaguars, and human health by explicitly warning hunters and local communities about the risk of Q fever.

The paper has a good methodology, and using diagnostic tools such as serology (ELISA) and molecular biology (PCR) provides a more complete picture.

Serological tests showed historical exposure, while PCR confirmed active infection.

The finding of a PCR-positive, seronegative animal is a classic and important demonstration of an acute infection.

The use of the IS1111 gene for *C. burnetii*-specific PCR is the gold standard and crucial for differentiating the pathogen from non-pathogenic *Coxiella*-like endosymbionts (CLE) found in ticks.

This adds high confidence to the single positive detection. The ethical aspects are important because they used animals previously hunted.

It provides scientific data without additional ethical costs and aligns the research with local practices.

The discussion effectively compares the findings (both seroprevalence and PCR prevalence) with those of other studies worldwide, providing good context.

Speculation on potential transmission routes (airborne, consumption of livestock carcasses/afterbirth) is well explained.

The authors have franked the limitations of the study. This is mainly due to the limited sample size.

This is the most significant limitation explicitly noted by the authors.

With only 36 animals, the confidence intervals for both seroprevalence (11.7%–38.1%) and PCR prevalence (0.1% – 14.5%) were very wide.

This means that the “true” infection rate in the population could be anywhere within these broad ranges and subdiagnostic.

The findings may indicate the circulation of *Coxiella*, but cannot be used to estimate its precise prevalence in the wider feral pig population.

The work is a cross-sectional design that shows a snapshot in time, and the study can demonstrate that infection is present, but cannot answer dynamic questions about how the pathogen is maintained and transmitted within the ecosystem.

It establishes an association, not causation. This is not a poor design and is not the author’s responsibility.

Hence, only one animal was PCR-positive out of the 36 tested.

While this is a crucial finding, it limits the ability to perform further analyses, such as genetic sequencing, to compare the strain with those found in livestock or humans in the region.

A larger sample size may yield more positive tissues for comparative studies.

Ticks were collected from only 4 four of the 36 animals (11%).

This convenience sampling means that the data obtained from *A. sculptum* ticks are not representative of the tick burden in the entire study population.

The statement that ticks may not be the main source of infection is uncautious because it is based on an extremely limited tick population.

The authors should be more cautious about this. While the discussion of alternative transmission routes (airborne, dietary) is appropriate, the study design does not allow for testing these hypotheses.

Again, the authors must be cautious regarding the limitations of the study.

In conclusion, this article is very interesting, especially for the intelligent way to obtain *Coxiella* from hunted animals.

The best contribution is to put a new host species and a critical region on the map for *C. burnetii*.

The investigators proved that *C. burnetii* is circulating in feral pigs in the Pantanal.

It effectively uses limited data to build a compelling case for why this matters and why larger, more longitudinal studies are urgently needed to understand the epidemiology and risk at this dynamic wildlife-livestock-human interface.

## AUTHORS’ RESPONSE TO THE REVIEWERS

We would like to thank the Editor and the Reviewers for their careful evaluation of our manuscript and for the constructive comments and suggestions.

We believe that these comments have significantly improved the quality and clarity of our manuscript.

Reviewer: 1

The manuscript under review describes the first detection of *Coxiella burnetii* in wild pigs (*Sus scrofa*) in Brazil, a finding of great relevance to One Health in the Pantanal.

The study is methodologically sound, especially in its use of specific molecular techniques to differentiate *C. burnetii* from related endosymbionts.

The originality of the work is very high, since this study represents the first detection of *C. burnetii* in free-living wild pigs in Brazil.

Below are the aspects that need improvement:

The abstract is well structured and informative, covering objectives, methods, results, and conclusion.

However, its adequacy can be improved through minor adjustments that reinforce the main narrative.

The abstract states that ‘Only *Coxiella*-like endosymbiont was detected in *Amblyomma sculptum* ticks.’ Include the percentage of infection.

Response: Thank you for the suggestion. We have included the percentage of infection at the end of the sentence, which now reads: “Only *Coxiella*-like endosymbiont was detected in *Amblyomma sculptum* ticks (9/23 - 39,13%; 95% CI: 22,2% – 59,2%)”.

The abstract states that the spleen, liver, and ticks were analysed by PCR.

The results indicate that only 26 of the 36 spleen and liver samples were analysed.

Although the difference is small, we suggest a slight rewriting of the abstract to indicate ‘Sera were tested by ELISA, and tissues (26 spleen and liver fragments) and tick samples...’

Response: Thank you for pointing this out. We understand the potential confusion.

In fact, serum samples were also analysed by PCR. We therefore rephrased the sentence in the abstract as follows: “Sera were tested by ELISA, and spleen, liver, sera and tick samples were analyzed by PCR.”

We also added the total number of tested animals in the following sentence: “Serological evidence of exposure was detected in 22.2% (8/36; 95% CI: 11,7% – 38,1%), while *C. burnetii* DNA was found in one spleen sample (1/36 - 2.8%; 95% CI: 0.1% – 14.5%).”

The study is pioneering, but the authors acknowledge the limitation of the small number of PCR-positive animals (only 1/36; 2.8%).

The importance of the molecular finding (the PCR-positive animal was seronegative) suggests an active infection.

However, it is necessary to explicitly emphasise, at the end of the Introduction or at the beginning of the discussion, that this low PCR detection rate means that the epidemiological role as an active reservoir should be interpreted with statistical caution (given the wide confidence interval).

Response: Thank you for the suggestion. We have added the following sentence in the discussion, in the context of the PCR positive:

The low PCR detection rate in means that the epidemiological role of feral pigs as an active reservoir should be interpreted with caution.”

The point about the seronegative but PCR-positive animal is essential.

The discussion should be expanded to explore possible biological and epidemiological explanations for this observation in the context of wild ungulates (e.g., acute infection prior to seroconversion, failure of immune response, or latent infection and localisation of the bacterium in the spleen).

Response: We agree and thank the reviewer for this important point. We have added the following sentence to the Discussion

“In fact, here *C. burnetii* DNA was amplified from a spleen sample of a seronegative animal (2,80%) - indicating active infection before antibody production (seroconversion), or immune evasion leading to latency and localization in the spleen - in a lower prevalence than described in wild boars from Spain (4.3% - 4/93) and slightly higher than the one reported in Italy (1.6% -1/63). (19,21)”

It is essential that the footnote in Supplementary Table 1 on ‘*only blood was collected*’ be explicit so that the reader immediately understands why tissue analysis (PCR) was performed on only 26 animals, rather than on all 36 animals.

Response: Thank you for this suggestion. To improve clarity, we revised the title of Supplementary Table 1 as follows:

“Suppl Table 1. Screening for *Coxiella burnetii* antibodies and DNA in 36 sera samples and 26 tissues of free-ranging feral pigs (*Sus scrofa*) and associated 23 ticks from Mato Grosso state, Brazil.”

Reviewer: 2

This is a valuable and important preliminary research study. It successfully achieved its primary goal of providing the first documented evidence of *Coxiella burnetii* exposure and infection in feral pigs in Brazil.

The study was well-designed as a field-based survey, used appropriate and validated diagnostic methods, and thoughtfully contextualized its findings within the One Health framework.

However, its main limitations are related to its scale and the inherent constraints of a cross-sectional study, which the authors have appropriately acknowledged.

This study fills a critical knowledge gap by demonstrating that an abundant invasive species, such as feral pigs, is exposed to and can carry *C. burnetii* in a key Brazilian biome is a significant finding.

This raises important concerns regarding wildlife health, livestock biosecurity, and public health.

The authors highlight the results in the context of One Health.

They connect it to livestock by mentioning the extensive cattle ranching in the Pantanal, wildlife by discussing potential spillover to predators like jaguars, and human health by explicitly warning hunters and local communities about the risk of Q fever.

The paper has a good methodology, and using diagnostic tools such as serology (ELISA) and molecular biology (PCR) provides a more complete picture.

Serological tests showed historical exposure, while PCR confirmed active infection.

The finding of a PCR-positive, seronegative animal is a classic and important demonstration of an acute infection.

The use of the IS1111 gene for *C. burnetii*-specific PCR is the gold standard and crucial for differentiating the pathogen from non-pathogenic *Coxiella*-like endosymbionts (CLE) found in ticks.

This adds high confidence to the single positive detection. The ethical aspects are important because they used animals previously hunted.

It provides scientific data without additional ethical costs and aligns the research with local practices.

The discussion effectively compares the findings (both seroprevalence and PCR prevalence) with those of other studies worldwide, providing good context.

Speculation on potential transmission routes (airborne, consumption of livestock carcasses/afterbirth) is well explained.

The authors have franked the limitations of the study. This is mainly due to the limited sample size.

This is the most significant limitation explicitly noted by the authors.

With only 36 animals, the confidence intervals for both seroprevalence (11.7%–38.1%) and PCR prevalence (0.1% – 14.5%) were very wide.

This means that the “true” infection rate in the population could be anywhere within these broad ranges and subdiagnostic.

The findings may indicate the circulation of *Coxiella*, but cannot be used to estimate its precise prevalence in the wider feral pig population.

The work is a cross-sectional design that shows a snapshot in time, and the study can demonstrate that infection is present, but cannot answer dynamic questions about how the pathogen is maintained and transmitted within the ecosystem.

It establishes an association, not causation. This is not a poor design and is not the author’s responsibility.

Hence, only one animal was PCR-positive out of the 36 tested.

While this is a crucial finding, it limits the ability to perform further analyses, such as genetic sequencing, to compare the strain with those found in livestock or humans in the region.

A larger sample size may yield more positive tissues for comparative studies.

Ticks were collected from only 4 four of the 36 animals (11%).

This convenience sampling means that the data obtained from *A. sculptum* ticks are not representative of the tick burden in the entire study population.

The statement that ticks may not be the main source of infection is uncautious because it is based on an extremely limited tick population.

The authors should be more cautious about this.

Response: Thank you for raising this important concern. We fully agree with the limitations highlighted.

We did not intend to rule out the role of ticks as a source of infection.

Based on our findings and previously published studies in Brazil, we revised the Discussion to adopt a more cautious interpretation.

The revised text now reads: “Although *A. sculptum* ticks have previously been found to be positive for *C. burnetii*, recent studies in feral pigs and associated ticks from São Paulo state - Brazil, no blood or tick samples showed to be positive in the qPCR for *C. burnetii* based on the IS1111 gene. (33,45) Indicating that ticks may not be the main source of infection for feral pigs in Brazil. *C. burnetii* is probably far more frequently transmitted through the airborne route or consumption than through tick bites.”

While the discussion of alternative transmission routes (airborne, dietary) is appropriate, the study design does not allow for testing these hypotheses.

Again, the authors must be cautious regarding the limitations of the study.

Response: We agree and appreciate this observation. To address this limitation, we added the following sentence to the Discussion:

“Additional studies involving feral pigs are needed to allow a better understanding of prevalence and transmission routes in Brazilian wetland ecosystems.”

---

## [Reviewer Report · REVIEWERS COMMENTS]

## REVIEWER #1

The revised version of the manuscript has been carefully evaluated.

The authors have adequately addressed all reviewers’ comments, and the quality of the manuscript has been significantly improved.

## REVIEWER #2

I have no comments; the authors followed the referees’ recommendations.